# Ultra-Hypofractionation for Whole-Breast Irradiation in Early Breast Cancer: Interim Analysis of a Prospective Study

**DOI:** 10.3390/biomedicines10102568

**Published:** 2022-10-13

**Authors:** Valeria Sigaudi, Micol Zannetti, Eleonora Ferrara, Irene Manfredda, Eleonora Mones, Gianfranco Loi, Marco Krengli, Pierfrancesco Franco

**Affiliations:** 1Department of Translational Medicine (DIMET), University of Eastern Piedmont, 28100 Novara, Italy; 2Department of Radiation Oncology, Maggiore della Carità University Hospital, 28100 Novara, Italy; 3Department of Medical Physics, Maggiore della Carità University Hospital, 28100 Novara, Italy

**Keywords:** early breast cancer, radiotherapy, hypofractionation, ultra-hypofractionation, breast-conserving surgery, radiation

## Abstract

We report on the early clinical outcomes of a prospective series of early breast cancer (EBC) patients treated with ultra-hypofractionated post-operative whole-breast irradiation (WBI) after breast-conserving surgery (BCS) and axillary management. Primary endpoints were patient’s compliance and acute toxicity. Secondary endpoints included physician-rated cosmesis and ipsilateral breast tumour recurrence (IBTR). Acute toxicity was evaluated at the end of WBI, 3 weeks and 6 months thereafter, according to the Common Terminology Criteria for Adverse Events (v. 5.0). Patients were treated between September 2021 and May 2022. The treatment schedule for WBI consisted of either 26 Gy in 5 fractions over one week (standard approach) or 28.5 Gy in 5 fractions over 5 weeks (reserved to elders). Inverse planned intensity-modulated radiation therapy (IMRT) was used employing a static technique. A total of 70 patients were treated. Fifty-nine were treated with the 26 Gy/5 fr/1 w and 11 with the 28.5 Gy/5 fr/5 ws schedule. Median age was 67 and 70 in the two groups. Most of the patients had left-sided tumours (53.2%) in the 26 Gy/5 fr/1 w or right-sided lesions (63.6%) in the 28.5 Gy/5 fr/5 ws group. Most of the patients had a clinical T1N0 disease and a pathological pT1pN0(sn) after surgery. Ductal invasive carcinoma was the most frequent histology. Luminal A intrinsic subtyping was most frequent. Most of the patients underwent BCS and sentinel lymph node biopsy and adjuvant endocrine therapy. All patients completed the treatment program as planned. Maximum detected acute skin toxicities were grade 2 erythema (6.7%), grade 2 induration (4.4%), and grade 2 skin colour changes. No early IBTR was observed. Ultra-hypofractionated WBI provides favourable compliance and early clinical outcomes in EBC after BCS in a real-world setting.

## 1. Introduction

Radiation therapy (RT) is a mainstay option in the multimodality management of breast cancer [1,2]. It is considered as a standard approach for early-stage breast cancer after breast-conserving surgery (BCS), resulting in a halving of the rates of any breast cancer recurrence at 10 years and a reduction by 1/6th of the breast-cancer-related mortality at 15 years. Post-mastectomy radiation therapy (PMRT) can reduce by more than 10% the rate of any recurrence at 10 years in node-positive women, leading to an 8% reduction in the 20-year breast cancer mortality rates [3,4]. For many decades, the standard RT schedule to deliver whole-breast irradiation (WBI) after BCS was 50 Gy in 2 Gy fractions over 5 weeks, followed by a 10–16 Gy boost dose to the lumpectomy cavity, given sequentially [5]. Based on the radiobiological features of breast cancer, hypofractionation (40–42.5 Gy in 15–16 fractions over 3 weeks) was then introduced and mature results of prospective randomized phase III trials demonstrated its (at least) equivalence in terms of both tumour control probability and normal tissue effects [6,7]. More recently, two prospective randomized studies, namely the FAST-Forward and FAST trials, tested the clinical efficacy of 5-fraction regimens (up to 26 Gy and 28.5 Gy, respectively) to deliver WBI over 1 or 5 weeks [8,9]. In particular, the FAST-Forward trial reported on clinical outcomes for WBI to 26 or 27 Gy delivered in 5 fractions over 1 week vs. 40 Gy in 15 fractions over 3 weeks in more than 4000 patients [8]. The 26 Gy in 5 fractions over 1 week regimen proved to be safe, effective, and simple to implement and became a new standard of care in this setting [10,11]. Five-fraction WBI allows for reduced overall treatment time, fewer treatment visits, decreased overall machine time, lower travel, and financial burden for the patient, together with lower costs to healthcare providers [12]. We herein report on the early results of our early clinical experience with 5-fraction WBI after BCS in a tertiary university hospital, through a prospective observational study.

## 2. Materials and Methods

A consecutive series of 70 breast cancer patients were treated with 5-fraction WBI after BCS at the Department of Radiation Oncology, ‘Maggiore della Carità University Hospital’ in Novara, Italy, from September 2021 to May 2022. Patients had a histologically proven diagnosis of breast adenocarcinoma, received prior BCS (lumpectomy or quadrantectomy) with negative surgical margins, and had pathological stage pT0–pT3, pN0–N1 according to the American Joint Committee-Union Internationale Contre le Cancer staging system (8th Edition 2017) [13]. Prior primary systemic therapy (PST), post-operative chemotherapy, target therapy, and endocrine therapy were allowed. All patients provided written informed consent with respect to the proposed treatment and clinical data management.

### 2.1. Setup, Simulation, and Target Definition

All patients were positioned on a wing board with both arms raised above the head and radiopaque markers placed along the clinical borders of the breast. Three-millimetre slice thickness axial images were acquired from the lower aspect of the mandible to the bases of the lung. The whole-breast clinical target volume (CTV) encompassed the palpable remaining glandular tissue of the breast, with a superior and inferior border comprised within the extent of the radiopaque markers, encompassing the palpable breast gland. The CTV was limited to a minimum distance of 5 mm from the skin surface and the thoracic wall. The whole-breast planning target volume (PTV) was generated by adding a 5 mm isotropic margin around the CTV and subsequently refined to account for the skin, thoracic surface, and organs at risk (OARs). The heart, ipsilateral and contralateral lungs, and contralateral breast were separately contoured OARs. The heart was outlined up to the level of the pulmonary trunk superiorly, with the inclusion of the pericardium and the exclusion of the major vessels. Other delineated OARs were the ipsilateral humeral head, the whole liver (right-sided cases), and the left anterior descending coronary artery (left-sided cases).

### 2.2. Radiotherapy Planning and Delivery

Patients were treated either with 26 Gy in 5 fractions over 1 week (26 Gy/5 fr/1 w) or with 28.5 Gy (28.5 Gy/5 fr/5 ws) in 5 fractions over 5 weeks. The standard of care for WBI after BCS was considered 26 Gy/5 fr/1 w, while 28.5 Gy/5 fr/5 ws was chosen for elderly and/or frail patients experiencing logistic issues and difficulties in reaching the radiation oncology department.

The treatments were planned with the Pinnacle v.16.b Treatment Planning Software, using a simple approach based on static step and shoot intensity-modulated tangential fields. The prescription was set to the median PTV dose according to the ICRU 83 guidelines. Dose distribution was optimized by inverse planning so that 95% of the PTVs received at least 95% of the prescription dose, minimizing occurrence of hot spots (i.e., Dmax < 105–107% of prescribed dose). Dose constraints for OARs were set to V_8Gy_ < 15%, (ipsilateral lung); V_2.5Gy_ < 10% (contralateral lung); V_7Gy_ < 5% and V_1.5Gy_ < 30% (heart); maximum dose (D_0.1cc_) < 1.6 Gy (contralateral breast); maximum dose (D_0.1cc_) < 15 Gy (left anterior descending artery (LAD)). Optimization was addressed to reduce both the mean lung dose (MLD) and mean heart dose (MHD) for ipsilateral lung and heart and the mean dose to ipsilateral humeral head at the lower achievable level. Excess irradiation (D_2cc_), defined as the percentage of the prescription dose delivered to a volume of 2 cc of normal tissues external to the PTV, was minimized. A 1.5 cm skin flash was used to optimize the beam fluences proximal to the patient surface to manage the dosimetric and geometric uncertainties due to the tissue–air interface. Intensity-modulated radiation therapy was delivered using 6 MegaVoltage photon beams generated by a Varian 600 DBX Linac equipped with a 120 leaf multi-leaf collimator Millennium. The surface-based image system Align RT was used for daily image guidance during the initial positioning and real-time monitoring during the treatment delivery setting a 5 mm intervention level for setup errors.

For setup verification purposes, tangential fields portal images were taken before the first treatment session and quantitatively compared with digitally reconstructed radiographs obtained from planning computed tomography.

### 2.3. Follow-Up, Toxicity, and Cosmesis

Patients were examined at the end of treatment, after 3 weeks and at 6 months from treatment. Afterward, a twice a year schedule was planned. Surveillance for disease recurrence at any site includes a clinical examination at every time point, planned chest X-ray and mammography once a year, and complete blood cell count and serological markers twice a year. Acute toxicity was assessed at the completion of treatment, at 3 weeks and after 3 months from WBI. The maximal detected toxicity was scored according to the Common Terminology Criteria for Adverse Events, version 5.0 [14]. Cosmetic results were assessed at the end of the radiation course and thereafter at 6 months of follow-up time, using a cosmetic evaluation scale consisting of different categories, namely excellent, good, fair, or poor, in agreement with the Harvard criteria [15]. At each follow-up examination, physicians were asked to judge the cosmetic results as follows: an ‘‘excellent’’ cosmetic result score was assigned when the treated breast looked essentially the same as the contralateral breast; a ‘‘good’’ cosmetic score was assigned for minimal but identifiable radiation effects of the treated breast; a ‘‘fair’’ score meant significant radiation effects were readily observable; a ‘‘poor’’ score was used for radiation-induced severe late effects of breast tissue.

## 3. Results

A total of 70 patients were enrolled in this prospective observational study. Fifty-nine patients were treated with the 26 Gy/5 fr/1 w schedule, while eleven were treated with the 28.5 Gy/5 fr/5 ws regimen.

### 3.1. Patients’ Characteristics

Patients treated with 26 Gy/5 fr/1 w had a mean age of 67 years, a median body mass index of 24.2, and a relatively low level of comorbid conditions and exposure to risk factors for treatment-related toxicity (diabetes: 6.8%; hypertension: 45.8%; vasculopathy: 16.9%; active smoking: 5.1%; alcohol intake: 0%). Patients in the 28.5 Gy/5 fr/5 ws group were older (mean age: 70 years) and with a higher rate of comorbid conditions and/or risk factors (diabetes: 18.2%; hypertension: 72.7%; vasculopathy: 27.3%; smoking: 9%). See Table 1 for details.

### 3.2. Tumour Characteristics

Breast tumours were mostly left-sided (53.2%) in the 26 Gy/5 fr/1 w group or right-sided (63.6%) in the 28.5 Gy/5 fr/5 ws. The most common breast localization was in the upper outer quadrant in both groups (41.9% for 26 Gy/5 fr/1 w and 27.3% for 28.5 Gy/5 fr/5 ws). Most of the patients had a clinical T1N0 disease stage at diagnosis (80.6% for 26 Gy/5 fr/1 w and 72.7% for 28.5 Gy/5 fr/5 ws) and a pathological pT1pN0-Nx(sn) stage after surgery (85.5% and 90.9%, respectively). Ductal invasive carcinoma was the most frequent histology (72.6% and 90.9%, respectively) and grade 1–2 the most frequent feature (80.6% and 63,6%, respectively). The most common intrinsic subtyping was Luminal A disease (58.1% and 45.5%, respectively). Vascular invasion (11.3% and 9.1%) and perineural invasion (17.7% and 18.2%) were infrequent. More details can be found in Table 2.

### 3.3. Treatment Characteristics

All the patients underwent BCS in both groups. The axilla was mostly managed conservatively. Post-operative systemic treatments mostly consisted of endocrine therapy (84.7% for 26 Gy/5 fr/1 w and 63.6% for 28.5 Gy/5 fr/5 ws). Adjuvant chemotherapy was given in 15.3% of patients in the 26 Gy/5 fr/1 w group, while trastuzumab was given in 13.6%. A total of 10.2% of patients in the 26 Gy/5 fr/1 w group received primary systemic therapy. See Table 3 for details.

### 3.4. Dosimetric Results

Mean PTV volume was 531.4 cc for the 26 Gy/5 fr/1 w group and 532.9 cc for the 28.5 Gy/5 fr/5 ws cohort. With respect to target coverage, mean D98 was 24.7 Gy and 24.9 Gy, while mean D2 was 26.7 Gy and 29.0 Gy in the two groups, respectively. Mean V95 was 96.5% and 96.6%, while V105 was 0.02% and 0.03%.

In terms of dose received by OARs, mean V8 to ipsilateral lung was 13.6% in the 26 Gy/5 fr/1 w group and 12.8% in the 28.5 Gy/5 fr/5 ws group. Mean ipsilateral lung dose was 3.2 Gy and 3.3 Gy in the two groups. The V7 to the heart was 3.5% and 3.8%, while the mean heart dose was 1.2 Gy and 1.4 Gy in the two groups. The average mean dose to the LAD was 3.7 Gy and 3.9 Gy, while the average maximal dose was 13.4 Gy and 13.8 Gy in the two groups. The maximal dose to the contralateral breast was 1.9 Gy and 1.8 Gy in the two groups. More details are available in Table 4.

### 3.5. Feasibility, Toxicity Profile, and Cosmesis

Treatment was feasible, since all patients completed the treatment program as planned.

In both fractionation groups, no grade 3 nor 4 toxicities were recorded.

In the 26 Gy/5 fr/1 w group, maximum detected acute toxicity was observed 3 weeks after treatment, with a 6.7% rate of grade 2 erythema, a 4.4% rate of grade 2 skin induration, and a 2.2% rate of grade 2 skin pigment alteration, telangiectasia, pruritus, and pain (on 45 patients evaluated). See Table 5 for details.

In the 28.5 Gy/5 fr/5 ws group, one patient (9.1%) experienced grade 2 erythema at the end of WBI. At 3 weeks from treatment end, out of 6 patients evaluated, grade 1 erythema and atrophy were experienced by 1 (16.7%), grade 1 oedema by 2 (33%), and grade 1 skin hyperpigmentation by 3 (50%). See Table 6 for details.

## 4. Discussion

Hypofractionation is the standard radiotherapy schedule in the treatment of breast cancer [16]. The recent consensus of the European Society for Radiotherapy and Oncology Advisory Committee in Radiation Oncology Practice stated that moderately hypofractionated radiotherapy can be offered to any breast cancer patient planned to undergo WBI, chest wall, and nodal irradiation, including those who received a previous breast reconstruction [16]. This is based on the long-term results of the START B and Canadian trials and the most recent findings of the Danish Breast Cancer Group (DBCG) and Chinese trials [17,18]. Recently, the 5-year results of the FAST-Forward trial have been published [8]. The investigators tested two different experimental schedules of 5 fractions given over 1 week, delivering a total nominal dose 26 Gy or 27 Gy, over the standard 40 Gy in 15 fractions given in 3 weeks, to assess the non-inferiority of the experimental schedules in terms of ipsilateral breast tumour recurrence (IBTR), the primary endpoint of the study, and the hypothesis of comparable rates for late adverse effects. The study enrolled 4096 patients, after either BCS or mastectomy (pT1–T3, pN0–N1) [8]. The acute toxicity profile was previously reported for the 5-fraction schedules, showing a low incidence rate of adverse events [19]. The study reported an estimated 5-year local recurrence incidence of 1.4% after 26 Gy, 1.7% after 27 Gy, and of 2.1% after 40 Gy in 15 fractions, showing non-inferiority of the 27 and 26 Gy schedules compared with moderate hypofractionation. The 26 Gy schedule was found to be similar and, therefore, comparably safe as the 40 Gy schedule, in terms of late normal tissue effects [8,9,10]. No differential effect for the 26 Gy/5 fr/1 w schedule, in terms of IBTR, was found with respect to age, tumour grade and size, nodal status, tumour bed boost, adjuvant chemotherapy, and intrinsic subtyping. Moreover, no differential effect was found, in terms of normal tissue effects in the breast or chest wall, for the 26 Gy/5 fr/1 w regimen, with respect to age, breast size, surgical deficit, tumour bed boost, or adjuvant chemotherapy [10]. Following the publication of the results of the FAST-Forward trial and considering the widespread uptake of the 26 Gy/5 fr/1 w schedule in the United Kingdom, the Royal College of Radiologists Professional Support and Standards Board updated their 2016 breast radiotherapy consensus statement, adopting 26 Gy/5 fr/1 w regimen for WBI and partial breast and chest wall radiotherapy as the standard regimen [11]. In 2020, the 10-year follow-up results of the FAST trial were also published [9]. The study tested two experimental schedules of 5 fractions (28.5 Gy or 30 Gy in 5 fractions) compared to 50 Gy in 5 fractions [9]. The FAST trial compared the three schedules with the 5-fraction regimens delivered once weekly over 5 weeks to remove any confounding effects due to differences in overall treatment time [20]. The primary endpoint was photographic assessment at 2 and 5 years [9]. The trial was primarily aimed at producing a reliable estimate of tumour and normal tissue sensitivity to dose/fractionation. The first results of the FAST trial produced an estimated value of the α/β ratio of 2.6 Gy, consistent with the START trials [21]. The study showed no statistically significant difference in terms of normal tissue effects between the 28.5 Gy/5 fr/5 ws schedule and the 50 Gy in 25 fractions over 5 weeks regimen. The 10-year IBTR rate reported in the trial was 1.7% for the 28.5 Gy/5 fr/5 ws schedule, even if the study was not powered for statistical analysis and only 11 events were reported in the three arms [9,20]. The once-weekly 28.5 Gy/5 fr/5 ws schedule can be considered as a suitable radiotherapy option to increase access to radiotherapy for patients who struggle to reach the department due to older age, frailty, comorbid conditions, or logistic issues [20,22].

During implementation of breast ultra-hypofractionation in our radiation oncology department, we prioritized standardization of our clinical practice to build internal consensus and increase homogeneity in the decision-making process together with safety, quality, and efficiency. We chose a specific setting of breast cancer patients to be offered 26 Gy/5 fr/1 w WBI, specifically those submitted to BCS, with low–intermediate risk invasive cancers (pT1-T3, pN0-1) or ductal carcinoma in situ (DCIS) resected with margins ≥2 mm. Both primary systemic therapy and adjuvant chemotherapy were allowed. No age constraint was introduced since no age-dependent differential effect was seen in the FAST-Forward trial. A more cautious approach was proposed for the following clinical situations: post-mastectomy radiotherapy, especially in reconstructed patients, nodal irradiation, and clinical settings with the need for a boost dose to the lumpectomy cavity. For those patients, moderate hypofractionation was maintained as the standard of care. When difficulties were reported by the patient to attend protracted regimens or daily radiotherapy fractionation schemes, the once-weekly 28.5 Gy/5 fr/5 ws regimen was proposed. The most common situations of patients offered the FAST fractionation were poor performance status, comorbid conditions, potential decreased compliance to treatment, lack of a caregiver or family/social support, and financial issues.

Both the 5-fraction schedules employed to deliver WBI after BCS proved to be safe and feasible in a real-word setting within an academic tertiary hospital in Italy. The acute toxicity profile was generally mild with respect to both skin and subcutaneous tissues. Cosmetic outcomes were generally good to excellent in most of the cases. In terms of planning and delivery, a good target coverage and a favourable sparing of normal tissue was obtained with a relatively simple technique, deliverable without specific technical complications. The 28.5 Gy/5 fr/5 ws schedule was particularly suitable for frail patients, increasing the equitable access to radiotherapy also for individuals who may have radiotherapy omitted because of a supposed poor compliance to treatment [23]. Long-term real-world data are needed to further corroborate this finding, particularly in the settings of nodal irradiation and post-mastectomy radiotherapy, especially in patients who underwent breast reconstruction. The integration of a tumour bed boost dose with 5-fraction WBI regimens is to be considered an interesting field of clinical investigation.

## Figures and Tables

**Table 1 biomedicines-10-02568-t001:** Patients characteristics.

Pts Characteristics	N° (%)	
	26Gy/5Fr/1w	28.5Gy/5Fr/5ws
*Age*		
<60 years	11 (18.6)	1 (9.0)
≥60 years	48 (81.4)	10 (91.0)
Median (range)	67 (46–83)	70 (47–88)
*BMI*		
Median (range)	24.2 (17.9–35.7)	24.4 (20.1–30.2)
*Diabetes*		
Yes	4 (6.8)	2 (18.2)
No	55 (93.2)	9 (81.8)
*Hypertension*		
Yes	27 (45.8)	8 (72.7)
No	32 (54.2)	3 (27.3)
*Vasculopathy*		
Yes	10 (16.9)	3 (27.3)
No	49 (83.1)	8 (72.7)
*Smoking status*		
Yes	3 (5.1)	1 (9.0)
No	56 (94.9)	10 (91.0)
*Regular alcohol intake*		
Yes	0 (0.0)	0 (0.0)
No	59 (100.0)	11 (100.0)

Legend: N°, number; pts, patients; Fr, fractions; w, week; ws, weeks; BMI, body mass index.

**Table 2 biomedicines-10-02568-t002:** Tumour characteristics.

Tumor Characteristics	N (%)	
	26Gy/5Fr/1w	28.5Gy/5Fr/5ws
*Laterality*		
Left-sided	33 (53.2)	4 (36.4)
Right-sided	29 (46.8)	7 (63.6)
*Quadrants*		
Upper inner quadrant	6 (9.7)	1 (9.1)
Lower inner quadrant	3 (4.8)	1 (9.1)
Upper outer quadrant	26 (41.9)	3 (27.3)
Lower outer quadrant	6 (9.7)	0 (0.0)
Accross inner quadrants	3 (4.8)	1 (9.1)
Accross outer quadrants	2 (3.2)	2 (18.2)
Accross upper quadrants	7 (11.3)	1 (9.1)
Accross lower quadrants	1 (1.6)	0 (0.0)
Central quadrant	5 (8.1)	1 (9.1)
Axillary tail involvement	3 (4.8)	1 (9.1)
*Clinical tumour stage*		
cTis	3 (4.8)	1 (9.1)
cT1a	3 (4.8)	1 (9.1)
cT1b	21 (33.9)	4 (36.7)
cT1c	23 (37.1)	2 (18.2)
cT2	12 (19.4)	3 (27.3)
*Clinical nodal stage*		
cN0	60 (96.8)	10 (90.9)
cN1	2 (3.2)	1 (9.1)
*Pathological tumour stage*		
pTis	2 (3.2)	1 (9.1)
ypT0	5 (8.1)	0 (0.0)
pT1mic	1 (1.6)	0 (0.0)
pT1a	1 (1.6)	3 (27.3)
ypT1a	1(1.6)	0 (0.0)
pT1b	16 (25.8)	1 (9.1)
pT1c	27 (43.5)	2 (18.2)
pT2	9 (14.6)	3 (27.3)
pT3	0 (0.0)	1 (9.1)
*Pathological nodal stage*		
ypN0	5 (8.1)	0 (0.0)
pN0	48 (77.4)	6 (54.5)
pN1	6 (9.7)	1 (9.1)
pNx	3 (4.8)	4 (36.4)
*Histology*		
Ductal	45 (72.6)	10 (90.9)
Lobular	5 (8.1)	1 (9.1)
Mixed ductal/lobular	2 (3.2)	0 (0.0)
Mucinous carcinoma	4 (6.5)	0 (0.0)
Papillar carcinoma	3 (4.8)	0 (0.0)
DCIS	3 (4.8)	0 (0.0)
*Grading*		
G1	8 (12.9)	1 (9.1)
G2	42 (67.7)	6 (54.5)
G3	12 (19.4)	4 (36.4)
*Estrogen receptor*		
Positive	55 (88.7)	10 (90.9)
Negative	6 (9.7)	1 (9.1)
NA	1 (1.6)	0 (0.0)
*Progesteron receptor*		
Positive	40 (64.5)	9 (81.8)
Negative	21 (33.9)	2 (18.2)
NA	1 (1.6)	0 (0.0)
*c-erbB2*		
Amplification	7 (11.3)	1 (9.1)
No amplification	51 (82.3)	9 (81.8)
NA	4 (6.4)	1 (9.1)
*Ki-67*		
<20%	36 (58.1)	5 (45.5)
20–40%	16 (25.8)	5 (45.5)
>40%	6 (9.7)	0 (0.0)
NA	4 (6.4)	1 (9.1)
*Vascular invasion*		
Positive	7 (11.3)	1 (9.1)
Negative	43 (69.4)	8 (72.7)
NA	12 (19.6)	2 (18.2)
*Perineural invasion*		
Positive	11 (17.7)	2 (18.2)
Negative	39 (62.9)	7 (63.6)
NA	12 (19.6)	2 (18.2)

Legend: N°, number; Fr, fractions; w, week; ws, weeks; NA, not available; DCIS, ductal carcinoma in situ.

**Table 3 biomedicines-10-02568-t003:** Treatment characteristics.

Tumor Characteristics	N (%)	
	26Gy/5Fr/1w	28.5Gy/5Fr/5ws
*Surgery*		
Quad/Lump	5 (8.5)	9 (81.8)
Quad/Lump + SLNB	54 (91.5)	0 (0.0)
Quad/Lump + SLNB + AD	0 (0.0)	2 (18.2)
*Surgical margins*		
Positive or ‘close’	0 (0.0)	0 (0.0)
Negative	54 (91.5)	11 (100.0)
NA	5 (8.5)	0 (0.0)
*Adjuvant endocrine therapy*	*50/59 (84.7)*	*7/11 (63.6)*
Tamoxifene	6 (12.0)	0 (0.0)
Aromatase inhibitors	44 (88.0)	6 (85.7)
LH-RH analogue + Exemestane	0 (0.0)	1 (14.3)
*Adjuvant systemic therapy*	*9/59 (15.3)*	*0/11 (0.0)*
EC + Weekly taxol	3 (33.3)	0 (0.0)
AC + Weekly taxol	1 (11.1)	0 (0.0)
TC + Weekly taxol	1 (11.1)	0 (0.0)
Weekly taxol	3 (33.3)	0 (0.0)
TC	1 (11.1)	0 (0.0)
*Primary systemic therapy*	*6/59 (10.2)*	*0/11 (0.0)*
EC + Weekly taxol	4 (66.7)	0 (0.0)
AC + Weekly taxol	1 (16.7)	0 (0.0)
CBDCA + Weekly taxol	1 (16.7)	0 (0.0)
*HER2-blockade*	*8/59 (13.6)*	*0/11 (0.0)*
Neoadjuvant	3 (37.5)	0 (0.0)
Adjuvant	8 (100.0)	0 (0.0)

Legend: N°, number; Fr, fractions; w, week; ws, weeks; NA, not available; SLNB, sentinel lymph node biopsy; AD, axillary dissection; LH-RH, luteinizing hormone-releasing hormone; EC, epirubicin-cyclophosphamide; AC, adriamycin-cyclophosphamide; TC, taxotere-cyclophosphamide; CBDCA, carboplatin; HER2, Human Epidermal Growth Factor Receptor 2.

**Table 4 biomedicines-10-02568-t004:** Dosimetric results.

**PTV**			
		**26Gy/5Fr/1w**	**28.5Gy/5Fr/5ws**
		Mean (SD)	Mean (SD)
	Volume (cc)	531.4 (275.5)	532.9 (269.6)
Whole breast	D_98_ (Gy)	24.7 (0.9)	24.9 (0.9)
	D_2_ (Gy)	26.7 (0.9)	29.0 (1.0)
	V_95_ (%)	96.5 (2.0)	96.6 (1.9)
	V_105_ (%)	0.02 (0.1)	0.03 (0.1)
	V_107_ (%)	0.0 (0.0)	0.0 (0.0)
**OARs**			
		**26Gy/5Fr/1w**	**28.5Gy/5Fr/5ws**
		Mean (SD)	Mean (SD)
Ipsilateral lung	V_4_ (%)	18.4 (4.4)	17.9 (4.7)
	V_8_ (%)	13.6 (3.3)	12.8 (3.6)
	V_16_ (%)	8.4 (2.6)	7.9 (2.7)
	D_max_ (Gy)	25.5 (1.5)	25.2 (2.1)
	MLD (Gy)	3.2 (1.7)	3.3 (1.7)
Controlateral lung	D_max_ (Gy)	1.63 (1.7)	1.4 (1.7)
	V_2.5_ (%)	0.02 (0.07)	0.02 (0.07)
	V_5_ (%)	0.0 (0.0)	0.0 (0.0)
Heart (left-sided tumors)	V_1.5_ (%)	13.1 (7.78)	14.4 (7.8)
	V_7_ (%)	3.5 (8.9)	3.8 (2.3)
	MHD (Gy)	1.2 (0.8)	1.4 (0.9)
	D_max_ (Gy)	22.2 (5.4)	22.7 (4.2)
LAD (left-sided tumor)	D_mean_	3.7 (2.3)	3.9 (2.2)
	D_max_	13.4 (6.5)	13.8 (6.1)
Liver (right-sided tumors)	V_2.5_ (%)	4.9 (4.9)	5.3 (5.2)
	V_5_ (%)	3.3 (3.3)	3.5 (3.4)
	V_10_ (%)	2.1 (2.4)	2.3 (2.5)
	D_mean_	0.9 (0.8)	0.9 (0,9)
	D_max_ (Gy)	20.3 (7.7)	20.4 (7.8)
Controlateral breast	D_max_ (Gy)	1.9 (1.8)	1.8 (1.8)
	D_mean_ (Gy)	0.2 (0.2)	0.2 (0.2)
Ipsilateral humeral head	D_max_ (Gy)	0.8 (2.2)	0.8 (2.3)
	D_mean_ (Gy)	0.2 (0.5)	0.2 (0.5)

Legend: Fr, fractions; w, week; ws, weeks; NA, not available; PTV, planning target volume; OARs, organs at risk; LAD, left anterior descending coronary artery; MLD, mean lung dose; MHD, mean heart dose.

**Table 5 biomedicines-10-02568-t005:** Toxicity profile and cosmesis for the 26 Gy in 5 fractions over 1 week schedule.

Parameter	Grade N (%)														
	*End of Treatment*					*3 Weeks after RT*					*6 Months after RT*				
	*G0*	*G1*	*G2*	*G3*	*G4–G5*	*G0*	*G1*	*G2*	*G3*	*G4–G5*	*G0*	*G1*	*G2*	*G3*	*G4–G5*
Erythema	47 (79.7)	11 (18.6)	1 (1.7)	0 (0.0)	0 (0.0)	31 (68.9)	11 (24.4)	3 (6.7)	0 (0.0)	0 (0.0)	11 (91.7)	1 (8.3)	0 (0.0)	0 (0.0)	0 (0.0)
Skin ulceration	59 (100.0)	0 (0.0)	0 (0.0)	0 (0.0)	0 (0.0)	45 (100.0)	0 (0.0)	0 (0.0)	0 (0.0)	0 (0.0)	12 (100.0)	0 (0.0)	0 (0.0)	0 (0.0)	0 (0.0)
Skin induration	47 (79.7)	11 (18.6)	1 (1.7)	0 (0.0)	0 (0.0)	32 (71.1)	11 (24.4)	2 (4.4)	0 (0.0)	0 (0.0)	4 (33.3)	6 (50.0)	2 (16.7)	0 (0.0)	0 (0.0)
Atrophy	59 (100.0)	0 (0.0)	0 (0.0)	0 (0.0)	0 (0.0)	43 (95.6)	2 (4.4)	0 (0.0)	0 (0.0)	0 (0.0)	11 (91.7)	1 (8.3)	0 (0)	0 (0.0)	0 (0.0)
Skin hyperpigmentation	59 (100.0)	0 (0.0)	0 (0.0)	0 (0.0)	0 (0.0)	26 (57.8)	18 (40.0)	1 (2.2)	0 (0.0)	0 (0.0)	8 (66.7)	4 (33.3)	0 (0)	0 (0.0)	0 (0.0)
Skin hypopigmentation	59 (100.0)	0 (0.0)	0 (0.0)	0 (0.0)	0 (0.0)	44 (97.8)	0 (0.0)	1 (2.2)	0 (0.0)	0 (0.0)	12 (100.0)	0 (0.0)	0 (0)	0 (0.0)	0 (0.0)
Teleangiectasiae	59 (100.0)	0 (0.0)	0 (0.0)	0 (0.0)	0 (0.0)	44 (97.8)	0 (0.0)	1 (2.2)	0 (0.0)	0 (0.0)	10 (83.3)	2 (16.7)	0 (0)	0 (0.0)	0 (0.0)
Oedema	56 (94.9)	3 (5.1)	0 (0.0)	0 (0.0)	0 (0.0)	40 (88.9)	5 (11.1)	0 (0.0)	0 (0.0)	0 (0.0)	12 (100.0)	0 (0.0)	0 (0)	0 (0.0)	0 (0.0)
Pruritus	56 (94.9)	2 (3.4)	1 (1.7)	0 (0.0)	0 (0.0)	34 (75.6)	10 (22.2)	1 (2.2)	0 (0.0)	0 (0.0)	12 (100.0)	0 (0.0)	0 (0)	0 (0.0)	0 (0.0)
Pain	55 (93.2)	4 (6.8)	0 (0.0)	0 (0.0)	0 (0.0)	36 (80.0)	8 (17.8)	1 (2.2)	0 (0.0)	0 (0.0)	11 (91.7)	0 (0.0)	1 (8.3)	0 (0.0)	0 (0.0)
**Cosmesis**															
Definition	*Poor*	*Fair*	*Good*	*Excellent*		*Poor*	*Fair*	*Good*	*Excellent*		*Poor*	*Fair*	*Good*	*Excellent*	
	0 (0.0)	1 (1.7)	23 (39.0)	35 (59.3)		0 (0.0)	1 (2.2)	14 (31.2)	30 (66.6)		0 (0.0)	1 (8.3)	4 (33.3)	7 (58.4)	

Legend: N°, number; RT, radiation therapy.

**Table 6 biomedicines-10-02568-t006:** Toxicity profile and cosmesis for the 28.5 Gy in 5 fractions over 5 weeks schedule.

Parameter	Grade N (%)														
	*End of Treatment*					*3 Weeks after RT*					*6 Months after RT*				
	*G0*	*G1*	*G2*	*G3*	*G4–G5*	*G0*	*G1*	*G2*	*G3*	*G4–G5*	*G0*	*G1*	*G2*	*G3*	*G4–G5*
Erythema	9 (81.8)	1 (9.1)	1 (9.1)	0 (0.0)	0 (0.0)	5 (83.3)	1 (16.7)	0 (0.0)	0 (0.0)	0 (0.0)	4 (80.0)	1 (20.0)	0 (0.0)	0 (0.0)	0 (0.0)
Skin ulceration	11 (100.0)	0 (0.0)	0 (0.0)	0 (0.0)	0 (0.0)	6 (100)	0 (0.0)	0 (0.0)	0 (0.0)	0 (0.0)	5 (100.0)	0 (0.0)	0 (0.0)	0 (0.0)	0 (0.0)
Skin induration	10 (90.9)	1 (9.1)	0 (0.0)	0 (0.0)	0 (0.0)	6 (100)	0 (0.0)	0 (0.0)	0 (0.0)	0 (0.0)	3 (60.0)	1 (20.0)	1 (20.0)	0 (0.0)	0 (0.0)
Atrophy	11 (100.0)	0 (0.0)	0 (0.0)	0 (0.0)	0 (0.0)	5 (83.3)	1 (16.7)	0 (0.0)	0 (0.0)	0 (0.0)	5 (100.0)	0 (0.0)	0 (0.0)	0 (0.0)	0 (0.0)
Skin hyperpigmentation	10 (90.9)	1 (9.1)	0 (0.0)	0 (0.0)	0 (0.0)	3 (50.0)	3 (50.0)	0 (0.0)	0 (0.0)	0 (0.0)	4 (80.0)	1 (20.0)	0 (0.0)	0 (0.0)	0 (0.0)
Skin hypopigmentation	11 (100.0)	0 (0.0)	0 (0.0)	0 (0.0)	0 (0.0)	6 (100)	0 (0.0)	0 (0.0)	0 (0.0)	0 (0.0)	5 (100.0)	0 (0.0)	0 (0.0)	0 (0.0)	0 (0.0)
Teleangiectasiae	11 (100.0)	0 (0.0)	0 (0.0)	0 (0.0)	0 (0.0)	6 (100)	0 (0.0)	0 (0.0)	0 (0.0)	0 (0.0)	4 (80.0)	1 (20.0)	0 (0.0)	0 (0.0)	0 (0.0)
Oedema	10 (90.9)	1 (9.1)	0 (0.0)	0 (0.0)	0 (0.0)	4 (66.7)	2 (33.3)	0 (0.0)	0 (0.0)	0 (0.0)	4 (80.0)	1 (20.0)	0 (0.0)	0 (0.0)	0 (0.0)
Pruritus	10 (90.9)	1 (9.1)	0 (0.0)	0 (0.0)	0 (0.0)	6 (100)	0 (0.0)	0 (0.0)	0 (0.0)	0 (0.0)	5 (100.0)	0 (0.0)	0 (0.0)	0 (0.0)	0 (0.0)
Pain	11 (100.0)	0 (0.0)	0 (0.0)	0 (0.0)	0 (0.0)	6 (100)	0 (0.0)	0 (0.0)	0 (0.0)	0 (0.0)	5 (100.0)	0 (0.0)	0 (0.0)	0 (0.0)	0 (0.0)
**Cosmesis**															
Definition	*Poor*	*Fair*	*Good*	*Excellent*		*Poor*	*Fair*	*Good*	*Excellent*		*Poor*	*Fair*	*Good*	*Excellent*	
	0 (0)	0 (0)	6 (54.5)	5 (45.5)		0 (0)	0 (0)	4 (66.7)	2 (33.3)		0 (0)	0 (0)	2 (40.0)	3 (60.0)	

Legend: N°, number; RT, radiation therapy.

## Data Availability

Data are available upon request by contacting the corresponding author.

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
