# Peer review of "Ultra-Hypofractionation for Whole-Breast Irradiation in Early Breast Cancer: Interim Analysis of a Prospective Study"

_biomedicines, 2022, doi:10.3390/biomedicines10102568_

Round 1

Reviewer 1 Report

The study by Sigaudi et al. analyzes an important question in treatment of breast cancer patients. The authors presented an investigation of a non-trivial approach to radiation therapy, that is, the exposure of the whole breast with low dose of X-rays. The study is well documented and deserves publication.

My concern is whether Biomedicine is the right journal for such a study. To me this journal is interested in molecular mechanisms that underlie the therapeutic modalities rather than on purely clinical works. The decision is at the discretion of the editor. This concern can be overcome if the authors provide a mechanistic rationale for their findings. Otherwise I suggest clinical cancer/radiation oncology journals instead. 

Author Response

We thank all the reviewers for their helpful comments, which allowed us to improve our manuscript. Please find hereby a point-by-point response to the comments received.

Reviewer.1

The study by Sigaudi et al. analyzes an important question in treatment of breast cancer patients. The authors presented an investigation of a non-trivial approach to radiation therapy, that is, the exposure of the whole breast with low dose of X-rays. The study is well documented and deserves publication.

Thank you for the positive comment on our manuscript.

My concern is whether Biomedicine is the right journal for such a study. To me this journal is interested in molecular mechanisms that underlie the therapeutic modalities rather than on purely clinical works. The decision is at the discretion of the editor. This concern can be overcome if the authors provide a mechanistic rationale for their findings. Otherwise I suggest clinical cancer/radiation oncology journals instead. 

Thanks for your comment. We understand the concerns. However, we submitted our manuscript to the Special Issue ‘The role of hypofractionated radiotherapy in modern oncology’ in Biomedicine. As stated in the announcement, one of the main areas of interest is: ‘novel indications for hypofractionated radiotherapy’, which we think is pertinent to the topic of our manuscript. Mover, we have been invited by the editors to submit the paper, whose abstract had been reviewed by the editors before submission. We hope all this make our submission target a reasonable choice.

Reviewer 2 Report

The authors present a prospective real-world cohort of breast tumor patients received extreme hypofractionated postopeartive breast irradiation. In this manuscript the early results are reported. A clear strangth of the paper is the precise reporing of radiation constraints and doses. Weeknesses are not-randomised design and very short follow-up.

Methods are well designed and very clear reported, with the exception of cosmesis. It remain unclear, whether the reported cosmesis results from patients' or phsicians' assessments came (Tables 5-6). Consider reporting the cosmesis results later on, together with late toxicity, as of their meaningfullness is limited at this very early follow-up. Furthermore, abstract is inconsistent with repoted results in term os assessment time of acute toxicities: end of treatment, 3 weeks and 6 months in the abstract vs. 

Please consider removing secondary (late) endpoints from the abstract, as theye are not reported in this paper. Maybe moving endpoints to methods section could be considered.

The (toxicity and feasibility) results are consistent to the literature. Conclusions section seems to be a template-text, it must be revised. (Maybe the last paragraph of the discussion was ought to be the conclusion section?)

Used CTCAE version is inconsitent reported. Version v6.0 - as mentioned in the abstract - has not yet been relesed by NCI. In the section methods (2.3) version v3.0 is mentioned. Cited was version v5.0 (citation nr. 14), although accession date as May 2022 - while assessments must have been permormed even before, as patients had been treated between Sept. 2021 and May 2022.  

In methods (section 2.2) is 26Gy/5fr/1w as standard declared, while 28.5/5fr/5ws scheme as an option for elderly/frail pts. Nevertheless, it remains unclear, how exactly were the patents assigned for the two different treatment schedules. As of patients characteristics (Tab.1) the only significant difference was hypertension between the two groups. Please consider discussing these issues.

Please consider reporting feasilibility in the results section, as announced in the abstract. ("All patients completed the treatment program as planned.")

Tables are too large and some of them confusing splitted on more pages. A thorough revision is recommended. Six patents received neoadjuvant chemotherapy - in the reported TNM stages this could be cleared as reporting ypT and ypN status for these patients. 

An abbreviation list might help non-radiation oncologists understandig radiation specific terms and abbreviations (e.g. PTV, cc, OAR, MHD, etc.).

Author Response

Reviewer.2

The authors present a prospective real-world cohort of breast tumor patients received extreme hypofractionated postopeartive breast irradiation. In this manuscript the early results are reported. A clear strangth of the paper is the precise reporing of radiation constraints and doses. Weeknesses are not-randomised design and very short follow-up.

Thanks for the pertinent comments on our study.

Methods are well designed and very clear reported, with the exception of cosmesis. It remain unclear, whether the reported cosmesis results from patients' or phsicians' assessments came (Tables 5-6).

This is a good point. Thank you for pointing this out. We reported on physician-reported cosmesis. This information has been added in the abstract and the full text has been modified accordingly.

Consider reporting the cosmesis results later on, together with late toxicity, as of their meaningfullness is limited at this very early follow-up.

We agree with the reviewer that the relevance of these outcomes could be limited at this early timepoint. However, we would rather keep the report as it is, since these data may be an indirect measure of the quality of the overall management of the patient, including surgery, oncoplastic procedure and radiation therapy. This could be of interest since we are reporting real-world data.

Furthermore, abstract is inconsistent with repoted results in term os assessment time of acute toxicities: end of treatment, 3 weeks and 6 months in the abstract vs. 

Thank you for pointing this out. The correct sequence is: end of treatment, 3 weeks and 6 months. We corrected the full text accordingly.

Please consider removing secondary (late) endpoints from the abstract, as theye are not reported in this paper. Maybe moving endpoints to methods section could be considered.

Thank you. Late endpoints were removed from the abstract. Endpoint were also moved in the M&M section of the abstract.

The (toxicity and feasibility) results are consistent to the literature. Conclusions section seems to be a template-text, it must be revised. (Maybe the last paragraph of the discussion was ought to be the conclusion section?)

Yes, thanks. Thanks for highlighting this. We eliminated the conclusion section which is not mandatory as per journal’s editorial rules. The last part of the discussion contains the conclusion statements.

Used CTCAE version is inconsitent reported. Version v6.0 - as mentioned in the abstract - has not yet been relesed by NCI. In the section methods (2.3) version v3.0 is mentioned. Cited was version v5.0 (citation nr. 14), although accession date as May 2022 - while assessments must have been permormed even before, as patients had been treated between Sept. 2021 and May 2022.  

Apologies for the inconsistencies and thanks for spotting this out. We corrected the errors following the suggestions. The version used for the present study is v 5.0 which was issued in 2017. Access date was changed into May 2021 in Ref 14.

In methods (section 2.2) is 26Gy/5fr/1w as standard declared, while 28.5/5fr/5ws scheme as an option for elderly/frail pts. Nevertheless, it remains unclear, how exactly were the patents assigned for the two different treatment schedules. As of patients characteristics (Tab.1) the only significant difference was hypertension between the two groups. Please consider discussing these issues.

This is a good point. Thanks for raising it. The standard treatment was 26Gy/5fr/1w. The 5-week schedule was offered to patients having logistic issue to reach the radiation oncology department. This has been explicitly stated in the main text.

Please consider reporting feasilibility in the results section, as announced in the abstract. ("All patients completed the treatment program as planned.")

Thank you. Feasibility was added in Paragraph 3.5.

Tables are too large and some of them confusing splitted on more pages. A thorough revision is recommended.

Tables have been revised and reduced in size as requested.

Six patents received neoadjuvant chemotherapy - in the reported TNM stages this could be cleared as reporting ypT and ypN status for these patients. 

This is correct. Thank you. Table 2 has been modified accordingly.

An abbreviation list might help non-radiation oncologists understanding radiation specific terms and abbreviations (e.g. PTV, cc, OAR, MHD, etc.).

Thanks for the input. We eliminated all the abbreviations which were not explicitly explained with a full text, to improve readability. Now all the abbreviations present in the full text are pre-declared ‘in extenso’. We hope this may be considered suitable.

Reviewer 3 Report

I am grateful for the opportunity to review manuscript ID: biomedicines-1943190, entitled “Ultra-hypofractionation for whole breast irradiation in early breast cancer.”

The authors report their findings of a prospective study of 70 patients treated with post-op WBI in a hypo-fractionated regimen of 5 fractions. The standard approach was 26Gy/5fr/1w (59 patients). Older and more frail patients were treated with a modified regimen of 28Gy/5fr/5w (11 patients). Treatments were planned for IMRT and delivered using a static technique. Primary endpoints for this study were patient compliance and acute toxicity. Secondary endpoints were late toxicity, cosmesis, and ipsilateral breast recurrence.

The authors essentially report implementing this regimen in their local practice following the 5 yr outcomes reported from the UK FAST Forward trial, but in doing so, prospectively tracking outcomes. The study design is straight forward, and the manuscript is well written. Results are clearly presented, and the conclusions are supported by the findings. I only have minor comments:

·        As the treatments reported on where recent, (Sept 2021 – May 2022), and late toxicity is listed as a secondary endpoint, perhaps revise the title to “Ultra-hypofractionation for whole breast irradiation in early breast cancer: Interim analysis of a prospective study”.

·        Some minor grammatical edits required throughout ahead of final publication.

Author Response

We thank all the reviewers for their helpful comments, which allowed us to improve our manuscript. Please find hereby a point-by-point response to the comments received.

Reviewer.3

I am grateful for the opportunity to review manuscript ID: biomedicines-1943190, entitled “Ultra-hypofractionation for whole breast irradiation in early breast cancer.”

The authors report their findings of a prospective study of 70 patients treated with post-op WBI in a hypo-fractionated regimen of 5 fractions. The standard approach was 26Gy/5fr/1w (59 patients). Older and more frail patients were treated with a modified regimen of 28Gy/5fr/5w (11 patients). Treatments were planned for IMRT and delivered using a static technique. Primary endpoints for this study were patient compliance and acute toxicity. Secondary endpoints were late toxicity, cosmesis, and ipsilateral breast recurrence.

The authors essentially report implementing this regimen in their local practice following the 5 yr outcomes reported from the UK FAST Forward trial, but in doing so, prospectively tracking outcomes. The study design is straight forward, and the manuscript is well written. Results are clearly presented, and the conclusions are supported by the findings.

Thank you for the positive comment on our manuscript.

 I only have minor comments:

As the treatments reported on where recent, (Sept 2021 – May 2022), and late toxicity is listed as a secondary endpoint, perhaps revise the title to “Ultra-hypofractionation for whole breast irradiation in early breast cancer: Interim analysis of a prospective study”.

Thank you for the suggestion. The title has been revised accordingly.

Some minor grammatical edits required throughout ahead of final publication.

Thank you. Grammar and spelling have been double-checked and revised where needed.

Round 2

Reviewer 1 Report

The authors convinced me that Biomedicines was the right submission. The study is now recommended for publication. 

Reviewer 2 Report

Thank you for revising the article. 

I suggest to accept it for publication in present form.